# Impact of axisymmetric deformation on MR elastography of a nonlinear tissue-mimicking material and implications in peri-tumour stiffness quantification

**Marco Fiorito**[1¤a]*, **Daniel Fovargue**[1], **Adela Capilnasiu**[1], **Myrianthi Hadjicharalambous**[1¤b], **David Nordsletten**[1,2], **Ralph Sinkus**[1,3], **Jack Lee**[1]*

**1** School of Biomedical Engineering and Imaging Sciences, King's College London, London, United Kingdom, **2** Department of Biomedical Engineering and Cardiac Surgery, University of Michigan, Ann Arbor, Michigan, United States of America, **3** U1148, INSERM, Hôpital Bichat, Paris, France

¤a Current address: Department of Biomedical Engineering, University of Basel, Allschwil, Switzerland
¤b Current address: Department of Mechanical Engineering, University of Cyprus, Nicosia, Cyprus
* fiorito.marco@hotmail.com (MF); jack.lee@kcl.ac.uk (JL)

**Data Availability Statement:** All data are available at www.osf.io (DOI 10.17605/OSF.IO/8TFNC).

## Abstract

Solid tumour growth is often associated with the accumulation of mechanical stresses acting on the surrounding host tissue. Due to tissue nonlinearity, the shear modulus of the peri-tumoural region inherits a signature from the tumour expansion which depends on multiple factors, including the soft tissue constitutive behaviour and its stress/strain state. Shear waves used in MR-elastography (MRE) sense the apparent change in shear modulus along their propagation direction, thereby probing the anisotropic stiffness field around the tumour. We developed an analytical framework for a heterogeneous shear modulus distribution using a thick-shelled sphere approximation of the tumour and soft tissue ensemble. A hyperelastic material (plastisol) was identified to validate the proposed theory in a phantom setting. A balloon-catheter connected to a pressure sensor was used to replicate the stress generated from tumour pressure and growth while MRE data were acquired. The shear modulus anisotropy retrieved from the reconstructed elastography data confirmed the analytically predicted patterns at various levels of inflation. An alternative measure, combining the generated deformation and the local wave direction and independent of the reconstruction strategy, was also proposed to correlate the analytical findings with the stretch probed by the waves. Overall, this work demonstrates that MRE in combination with non-linear mechanics, is able to identify the apparent shear modulus variation arising from the strain generated by a growth within tissue, such as an idealised model of tumour. Investigation in real tissue represents the next step to further investigate the implications of endogenous forces in tissue characterisation through MRE.

**Funding:** This research was funded by the European Union's Horizon 2020 Research and Innovation Programme (ec.europa.eu/programmes/horizon2020) under grant agreement No 668039 (RS, JL, DN). The authors would like to acknowledge funding from the King's College London & Imperial College London EPSRC Centre for Doctoral Training in Medical Imaging (EP/L015226/1) (www.imagingcdt.com). The research was funded/supported by the National Institute for Health Research (NIHR) Biomedical Research Centre based at Guy's and St Thomas' NHS Foundation Trust and King's College London (www.nihr.ac.uk). The views expressed are those of the author(s) and not necessarily those of the NHS, the NIHR or the Department of Health.

**Competing interests:** The authors have declared that no competing interests exist.

## Introduction

Solid tumour growth is often associated with an increase of mechanical stresses acting on the surrounding host tissue. Enhanced intra- and peri-tumoural stresses are mechanical indicators of solid tumour progression [1] and directly result in the deformation of the typically softer host tissues. Tumour-generated stress exerted on the adjacent soft tissue arises from two different major sources:

- *solid*: cancer cell proliferation associated to tumour progression [2], electrostatic repulsive forces generated among closely spaced hyaluronan chains [3] and the resistance of the host tissue to the deformation generated by the growing tumour mass [4] define the solid-phase component of the total stress;

- *fluid*: elevated tumour interstitial fluid pressure (IFP), caused by leaky or non-functional vessels and poor interstitial fluid drainage, leads to the accumulation of a fluid stress [5].

Solid stress presents two directionality-dependent components: a radial and a circumferential component. While the former acts compressively and gradually decreases when moving away from the tumour rim, the latter turns from compressive to tensile at the soft tissue interface with the tumour [6]. Mathematical models and experimental data have instead shown that IFP is responsible for compressive stress and uniformly distributed throughout the tumour core. These two components add up to define a total stress, which simultaneously compresses the tumour core and pushes against the host tissue. However, IFP quickly drops to zero at the periphery of the tumour, making solid stress the sole responsible for host tissue displacement [7, 8]. The strain generated by a pressurised tumour can shift the modulus of the surrounding host tissue, due to the nonlinear stress-strain behaviour typically exhibited by soft tissue when subjected to large-scale deformations [9]. The amplitude and distribution of the generated variation in elastic properties will depends on the tumour shape, the magnitude and direction of the generated deformation field, as well as on the underlying nonlinearity of the tissue.

Altered mechanical properties of soft tissue can be useful in diagnosing pathological conditions and cancer. A significant example is given by manual palpation in the case of breast cancer, a recommended practice for the detection of hard masses which was found to lead to the early diagnosis of more than 50% of asymptomatic cancers [10]. Exploiting the same principle, magnetic resonance elastography (MRE) has emerged in recent years as a valuable imaging technique capable of characterising the biomechanical properties of basically every organ [11–22]. Such technique relies on the processing of MR images of the harmonic micro-deformations generated by propagating shear waves, to reconstruct a shear modulus map of the imaged tissue. Several studies have shown the possibility to differentiate between healthy tissue, benign cancers and malignant lesions based on their stiffness using MRE [23–31]. However, all these results neglect the presence of external or endogenous forces, in order to satisfy the linear viscoelasticity approximation usually made for the shear modulus reconstruction. Given the nonlinear nature of soft tissue, this can lead to a bias in the material properties estimation [32]. Only recently have some MRE reconstruction methods begun to account for deformation of the tissue in response to endogenous or external forces [33, 34]. Under the simplifying assumption of uniaxial compression, previous mathematical analyses have shown that, by using a linear viscoelasticity assumption in an underlying nonlinear material, the probed variation in shear modulus depends on the direction of propagation of the shear waves with respect to the local deformation field [33]. Following such findings, the peri-tumoural habitat is expected to inherit a displaced pattern of elastic modulus due to the forces exerted on it by the pressurised tumour, which quantitatively correlates to the total stress distribution.

Elevated solid stress has recently emerged as a marker of tumour growth, and it has been associated to tumour invasion and metastasis [35–37], making its quantification crucial. While a few *ex vivo* measuring methods have been proposed [2, 38], up to now *in vivo* solid stress quantification remains an ongoing challenge. Nieskovski *et al.* (2017) have recently presented a novel technique based on a modified piezoelectric pressure catheter [39], however its invasive nature pushes towards the search for alternatives quantification methods. A recent paper by Fovargue *et al.* (2020) presented a proof-of-concept method to non-invasively infer the solid stress generated by a tumour by embedding a nonlinear mechanical model into MRE reconstruction [40]. Understanding the underlying relation between mechanical loading and evolution of an apparent elasticity tensor can allow a direct association between local variation in stiffness and tumour growth, and could open up an avenue for non-invasive retrieval of the underlying tumour-generated stress through an inverse approach.

In this paper we present a mathematical framework that models the altered shear modulus in the host tissue in response to the pressure generated by an idealised growing tumour, as measured by MRE. Starting from the formulation proposed by [33], the analytical model was extended to the case of an expanding thick-shelled sphere and was used to predict the resulting apparent anisotropic variation of the modulus. Experimental phantom results confirmed the validity of the developed formulation, offering a way to bridge such analytical considerations to an *in vivo* setting. This paper, combined with the results presented by Fovargue *et al.* (2020) [40], provides the foundations for the development of a quantitative diagnostic approach based on the integration of MRE data and tissue nonlinear mechanics.

## Analytical model

The mechanical properties of soft tissue determine the propagation of the periodic waves employed in elastography. Quantification, through MR imaging, of the micro-deformation produced by the probing shear waves enables to estimate the shear modulus of the investigated tissue. It is well established that a macroscopic deformation applied to a nonlinear viscoelastic material leads to a variation of its mechanical properties; however, its impact on the additional small-scale displacement produced by the harmonic waves is not obvious but can be determined through scale separations. In the first section, a set of equations of motion that captures the combined viscoelastic effects of a large-strain and small-scale harmonic motion propagation, developed in [33], is presented. Supporting information on the basic concepts in continuum mechanics could be found in standard texts [41]. This framework is then applied to the case of the radial inflation of a thick-shelled sphere, an idealisation of the tumour and soft tissue ensemble. Analytical quantification of the shear modulus anisotropic distribution around the inner sphere is also presented, using a modified Mooney-Rivlin constitutive equation. Finally, an analytical method to retrieve the shear modulus of the undeformed object from the material parameters used to model its hyperelastic behaviour is presented.

### Linearised constitutive laws of a large-strain viscoelastic body subjected to small perturbations

The application of an external force onto a deformable incompressible solid, defined in the reference configuration $\Omega_0 \subset \mathbb{R}^3$, leads to the diffeomorphic mapping of each point from their reference position $X \in \Omega_0$ into a new location $x(X, t)$. This new location is defined in the current domain $\Omega \subset \mathbb{R}^3$ at time $t \in [0, T]$ through the Lagrangian displacement $U(X, t) = x(X, t) - X$. Considering periodic waves acting in a time domain $I \subset [0, T]$ and under the

assumption of steady state, the generated micro-deformation and associated hydrostatic pressure

$$\boldsymbol{u}_\varepsilon(\boldsymbol{x}, t) = \mathrm{Re}\{\boldsymbol{u}_\mathbb{C}(\boldsymbol{x})e^{i\omega t}\} \qquad \text{and} \qquad p_\varepsilon(\boldsymbol{x}, t) = \mathrm{Re}\{p_\mathbb{C}(\boldsymbol{x})e^{i\omega t}\} \tag{1}$$

are complex-valued functions of space, where $\boldsymbol{u}_\mathbb{C}(\boldsymbol{x}) = \boldsymbol{u}_r(\boldsymbol{x}) + i\boldsymbol{u}_i(\boldsymbol{x})$ and $p_\mathbb{C}(\boldsymbol{x}) = p_r(\boldsymbol{x}) + ip_i(\boldsymbol{x})$. Given the scale separation, we can consider the low-amplitude harmonic waves $\boldsymbol{u}_\varepsilon$ as a perturbation of the pre-applied macro-deformation $\boldsymbol{U}$. The two deformation fields and their associated hydrostatic pressures can then be linearly combined as follows

$$\boldsymbol{U}^\varepsilon = \boldsymbol{U} + \boldsymbol{u}_\varepsilon \tag{2a}$$

$$P^\varepsilon = P + p_\varepsilon \tag{2b}$$

Following the formulation proposed by [33], the Eulerian form of the perturbed version of the equations of motion describing the propagation of elastic waves through a macroscopically deformed viscoelastic object with density $\rho$ can be written as:

$$\rho\omega^2\boldsymbol{u}_\mathbb{C} + \nabla_{\boldsymbol{x}} \cdot ((\boldsymbol{\mathcal{G}}' + i\boldsymbol{\mathcal{G}}'') : \nabla_{\boldsymbol{x}}\boldsymbol{u}_\mathbb{C} + p_\mathbb{C}\mathbb{1}) = 0 \tag{3a}$$

$$\nabla_{\boldsymbol{x}} \cdot \boldsymbol{u}_\mathbb{C} = 0 \tag{3b}$$

Eq 3b indicates the incompressibility of the material subjected to the micro-deformation $\boldsymbol{u}_\mathbb{C}$, while Eq 3a determines the dynamic behaviour of the propagating shear waves through the real and imaginary components of the complex viscoelasticity tensor, $\boldsymbol{\mathcal{G}}^* = \boldsymbol{\mathcal{G}}' + i\boldsymbol{\mathcal{G}}''$, defined as

$$\boldsymbol{\mathcal{G}}' = \frac{1}{J}\nabla_{\boldsymbol{F}}\boldsymbol{P}\boldsymbol{F}^T\boldsymbol{F}^T \qquad \text{and} \qquad \boldsymbol{\mathcal{G}}'' = \frac{\omega}{J}\nabla_{\frac{\partial \boldsymbol{F}}{\partial t}}\boldsymbol{P}\boldsymbol{F}^T\boldsymbol{F}^T \tag{4}$$

Here, $\boldsymbol{F}$ is the deformation gradient, relating the particle positions before and after the deformation, with $J$ being its Jacobian. $\boldsymbol{P}$ is the first Piola-Kirchhoff (PK1) stress tensor. Further details on the derivation of the presented equations of motion can be found in the S1 Appendix.

## Wave propagation in expanded thick-shelled sphere

In this manuscript, we have idealised the tumour-soft tissue ensemble as a thick-shelled sphere, with the inner void sphere representing the expanding tumour and exerting an isotropic pressure onto the outer sphere, which represents the surrounding host tissue. Following the formulation proposed by [42], we denote with $\boldsymbol{X} = \boldsymbol{X}(R, \Theta, \Phi)$ and $\boldsymbol{x} = \boldsymbol{x}(r, \theta, \varphi)$ the location of a material particle in spherical coordinates before and after the macro-deformation, where

$$A \leq R \leq B \qquad 0 \leq \Theta \leq 2\pi \qquad 0 \leq \Phi \leq \pi$$
$$a \leq r \leq b \qquad 0 \leq \theta \leq 2\pi \qquad 0 \leq \varphi \leq \pi.$$

Here, $A$ and $B$ are the radii of the inner and outer sphere in its undeformed state, while $a$ and $b$ are the corresponding radii after the expansion. A spherical deformation state can then be

described by the following tensors

$$\boldsymbol{F} = \begin{bmatrix} \dfrac{R^2}{r^2} & & \\ & \dfrac{r}{R} & \\ & & \dfrac{r}{R} \end{bmatrix} \qquad \boldsymbol{C} = \boldsymbol{F}^T\boldsymbol{F} = \begin{bmatrix} \dfrac{R^4}{r^4} & & \\ & \dfrac{r^2}{R^2} & \\ & & \dfrac{r^2}{R^2} \end{bmatrix} \qquad \boldsymbol{C}^2 = \begin{bmatrix} \dfrac{R^8}{r^8} & & \\ & \dfrac{r^4}{R^4} & \\ & & \dfrac{r^4}{R^4} \end{bmatrix} \tag{5}$$

where $r$ is a pure function of $R$ at each point and $\theta = \Theta$ and $\varphi = \Phi$ due to symmetry.

In the case of a hyperelastic material, the mechanical response of the outer sphere can be modelled through a strain energy density function $W = W(I_{\mathbf{C}}, II_{\mathbf{C}}, III_{\mathbf{C}})$, which, in light of the isotropy assumption, is a direct function of the invariants of the right Cauchy-Green stress tensor $\boldsymbol{C}$. The PK1 stress tensor can then be calculated using the following relation: $\boldsymbol{P} = \partial W/\partial \boldsymbol{F}$. Through tensor decomposition, we can express $\boldsymbol{P}$ as

$$\boldsymbol{P} = \frac{\partial W(I_{\hat{\mathbf{C}}}, II_{\hat{\mathbf{C}}})}{\partial \boldsymbol{F}} + PJ\boldsymbol{F}^{-T} \tag{6}$$

with the first term containing an incompressible strain energy density function, dependent on the unimodular invariants $I_{\hat{\mathbf{C}}} = J^{-\frac{2}{3}}I_{\mathbf{C}}$ and $II_{\hat{\mathbf{C}}} = J^{-\frac{4}{3}}II_{\mathbf{C}}$, and the second containing the hydrostatic pressure $P$. Application of the chain rule to the derivative leads to the following expression:

$$\boldsymbol{P} = -2\underbrace{\left[\frac{\partial W}{\partial I_{\hat{\mathbf{C}}}}\frac{I_{\hat{\mathbf{C}}}}{3} + \frac{\partial W}{\partial II_{\hat{\mathbf{C}}}}\frac{2II_{\hat{\mathbf{C}}}}{3}\right]\boldsymbol{F}^{-T}}_{①} \quad +2J^{-\frac{2}{3}}\underbrace{\left[\frac{\partial W}{\partial I_{\hat{\mathbf{C}}}} + \frac{\partial W}{\partial II_{\hat{\mathbf{C}}}}I_{\hat{\mathbf{C}}}\right]\boldsymbol{F}}_{②}$$
$$- 2\underbrace{\left[\frac{\partial W}{\partial II_{\hat{\mathbf{C}}}}J^{-\frac{4}{3}}\boldsymbol{F}\boldsymbol{C}\right]}_{③} + PJ\boldsymbol{F}^{-T}. \tag{7}$$

The radial stress component in the current configuration is defined as the inflating pressure $p_i$ (i.e. solid stress) exerted on the inner surface of the sphere. This can be evaluated by substituting the Cauchy stress tensor, $\boldsymbol{\sigma} = J^{-1}\boldsymbol{P}\boldsymbol{F}^T$, into the Cauchy momentum equation at equilibrium in the spherical coordinate system and integrating between $r \in [a, b]$. Since the stress generated by the expanding inner sphere decreases with radial distance, if we assume that $r = b$ is far from the centre, then the stress would approach 0. In the presence of alternative data at the boundary, this value can be substituted in and the formulation can be adapted accordingly. As such data is absent, in this work we assumed $\sigma_{rr}(r = a) = p_i$ and $\sigma_{rr}(r = b) = 0$, from which we obtain

$$p_i = \int_a^b \frac{\mathrm{d}\sigma_{rr}}{\mathrm{d}r}\,\mathrm{d}r \qquad \text{with} \qquad \frac{\mathrm{d}\sigma_{rr}}{\mathrm{d}r} = 2\frac{\sigma_{\theta\theta} - \sigma_{rr}}{r} \tag{8}$$

Assuming a specific strain energy density function, an analytical expression of the inflating pressure as a function of the radial stretch can be found. Considering a steady-state where transient material relaxation has ceased, we choose to model the rheology of the deformed thick-shelled sphere using a second order polynomial constitutive equation based on the Mooney-Rivlin model, under the simplifying assumption of a purely elastic material:

$$W = \frac{1}{2}\mu_1(I_{\hat{\mathbf{C}}} - 3) + \frac{1}{2}\mu_2(II_{\hat{\mathbf{C}}} - 3)^2 \tag{9}$$

where $\mu_1$ and $\mu_2$ are the material parameters. This model was previously employed in [33] to validate the impact of large scale deformations on the estimation of the mechanical properties in PVA samples. Given the roughly incompressible nature of most soft tissues, no volumetric energy component was added to the employed strain energy density function [9, 43]. The resulting inflating pressure reads

$$
p_i = \frac{R}{5r^4} \left[ -5(a^3 - A^3)(\mu_1 + 4\mu_2) + 25(\mu_1 + 4\mu_2)r^3 \right.
$$
$$
\left. -8\mu_2 \frac{(a^3 - A^3)r^6}{R^6} - 28\mu_2 \frac{r^6}{R^3} + 60\mu_2 \frac{r^2(a^3 - A^3 + r^3)}{R^2} \right]_{r=a}^{r=b}
$$

(10)

where $R = (A^3 - a^3 + r^3)^{1/3}$ following the incompressibility assumption.

Substituting Eq 9 in Eq 7, we can also evaluate the apparent variation in the elasticity tensor $\mathcal{G}'$ (see Eq 4) caused by the macro-deformation $U$. The derivatives of the three terms in brackets in Eq 7, i.e.: ① ② and ③, are given by the following equations:

$$
\frac{\partial①}{\partial F_{kn}} = \frac{1}{3}J^{-\frac{2}{3}}[\mu_1 + 4\mu_2 I_{\hat{C}}(2II_{\hat{C}} - 3)]F_{kn} - \frac{4}{3}J^{-\frac{4}{3}}\mu_2(2II_{\hat{C}} - 3)F_{nl}F_{ln}F_{nk}
$$
$$
-\frac{1}{9}[\mu_1 I_{\hat{C}} + 8\mu_2 II_{\hat{C}}(2II_{\hat{C}} - 3)]F_{nk}^{-1}
$$

(11a)

$$
\frac{\partial②}{\partial F_{kn}} = 2J^{-\frac{4}{3}}\mu_2(I_{\hat{C}}^2 + II_{\hat{C}} - 3)F_{kn} - 2J^{-2}\mu_2 I_{\hat{C}}F_{nl}F_{ln}F_{nk}
$$
$$
-\frac{1}{3}J^{-\frac{2}{3}}[\mu_1 + 6\mu_2 I_{\hat{C}}(II_{\hat{C}} - 1)]F_{nk}^{-1}
$$

(11b)

$$
\frac{\partial③}{\partial F_{kn}} = 2J^{-2}\mu_2 \left[ I_{\hat{C}}F_{kn} - J^{-\frac{2}{3}}F_{nl}F_{ln}F_{nk} - \frac{2}{3}J^{\frac{2}{3}}\left[II_{\hat{C}}\left(J^{-\frac{4}{3}} - 1\right) - 3\right]F_{nk}^{-1} \right](FC)_{is}
$$
$$
+J^{-\frac{4}{3}}\mu_2(II_{\hat{C}} - 3)\frac{\partial(FC)_{is}}{\partial F_{kn}}.
$$

(11c)

In order to obtain a complete expression for the PK1 stress tensor, the hydrostatic pressure must also be calculated. This can be derived by comparing the *rr* component of the Cauchy stress tensor, defined as a function of $P$ expressed in Eq 7, with the definition given in Eq 8, yielding

$$
P = \mu_1 \frac{R^6(-10r^6 + 9r^4R^2 + R^6)}{6r^4R^8}
$$
$$
+\mu_2 \frac{(r - R)^2(r + R)^2(r^2 + 2R^2)(5r^6 + 9r^2R^4 + 34R^6)}{6r^4R^8}.
$$

(12)

Using these results, finally the analytic expressions for $\mathcal{G}'$ and the stress-like tensor $\mathcal{G}' : \nabla_x \boldsymbol{u}_\varepsilon$ previously introduced can be derived. The double contraction between the stiffness tensor and the gradient of the wave-generated micro-deformation quantifies the apparent shear modulus sensed by the probing waves. In the simple case of plane waves, only one component of second-order tensor $\mathcal{G}' : \nabla_x \boldsymbol{u}_\varepsilon$ is non-zero and is identified by the scalar $G'$. The apparent variation in $G'$ caused by a spherical macroscopic compression will be presented in the Results.

## Intrinsic shear modulus

Estimation of the parameters characterising a hyperelastic material allows to retrieve its *intrinsic* shear modulus $\mu$ in the absence of deformation. Considering a simple shear deformation $x_1$

$= X_1 + \beta X_2$, where $\beta \ll 1$ under the limit of small strain, $\mu$ can be modelled as [44]:

$$\mu = \lim_{\beta \to 0} 2 \left( \frac{\partial W}{\partial I_{\hat{C}}} + \frac{\partial W}{\partial II_{\hat{C}}} \right) \tag{13}$$

Assuming a strain energy density function in the form presented in Eq 9, the intrinsic material shear modulus is calculated as

$$\mu = 2\mu_1. \tag{14}$$

## Materials and methods

### Plastisol phantom to mimic soft tissue nonlinear behaviour

To validate the developed mathematical framework, we have built an experimental set-up made of an inflatable balloon/catheter inserted into a cuboid phantom, characterised by a non-linear viscoelastic behaviour. In the next section, the details of the phantom construction are given. A uniaxial harmonic loading was carried out to identify the rheological behaviour of the material subjected to harmonic stress, as presented in the following section. Afterwards, a mathematical model to describe the viscoelastic response of tested material samples is given, while the last section explains the fitting process employed to assess its suitability to predict the experimental rheological data.

**Phantom construction.** The phantoms prepared for the inflation experiment were made with 80% soft-grade plastisol (Lure-solutions, UK) and 20% additional softener (non-phthalate plasticiser), following the protocols proposed by [45, 46]. 700 mL of the mixture were constantly stirred and heated up to $\sim 170°$C. The solution was then poured into a $86 \times 86 \times 115$ mm$^3$ tin cuboidal mould (Tinware Direct Ltd, Bedford, UK) and left to solidify at room temperature for a minimum of 12 h. 1% w/v silica particles were homogeneously scattered into the mixture when the material was still in a semi-liquid state, to help track the deformation field from the analysis of the MR images. Compression tests from a different study demonstrated that the addition of 1% w/v of micro-sized beads had little or no impact on the intrinsic elastic modulus of plastisol samples [47]. Once gently removed from the mould, the meniscus formed on the open side of the mould, caused by a quicker polymerisation around the walls, was cut off to preserve the cuboid geometry, leaving an approximately $86 \times 86 \times 70$ mm$^3$ cube.

**Rheological characterisation of plastisol material.** To assess the viscoelastic behaviour of the phantom, a rheological characterisation of the material was carried out at the Institute of Bioengineering of Queen Mary University of London (London, UK). Material samples were subjected to uniaxial harmonic micro-compressions, assumed to mimic the conditions generated by the propagating waves used in MRE. Additional static macro-compressions were superimposed to the periodic loading. In detail, each sample was placed between the parallel plates of a BOSE Electroforce 5500 controlled stress rheometer, ensuring that both plates were fully in contact with the cuboid. This instrument allowed for a compression in the vertical direction ($\sigma_{(3,3)} = \sigma$), while keeping the other directions stress-free ($\sigma_{(1,1)} = \sigma_{(2,2)} = 0$), and for the simultaneous measurement of the displacement of the top plate. All samples were preconditioned through an initial 13% compression for 5 min. Each sample was then re-positioned between the plates, compressed along the vertical direction and subjected to six cycles of 0.5mm ($\sim 1.5\%$) harmonic micro-deformations along the same direction, at increasing frequencies for a pre-determined period. This procedure was repeated for four different levels of compression (7%, 13%, 20%, 26%) while alternating the samples. Afterwards, each sample was re-positioned and a 4 mm ($\sim 11\%$) harmonic macro-deformation was applied around a mean

**Table 1. Details of the protocol used for the rheological test.** Either harmonic micro- ($\sim 1.5\%$) or macro-deformation ($\sim 11\%$) was applied on top of the uni-axial compression of the samples.

| | Uniax.-compression | Frequency (Hz) | 0.1 | 0.5 | 1 | 2 | 5 | 10 |
|---|---|---|---|---|---|---|---|---|
| **Micro-oscill.** ±1.5% | 7% | **Duration (s)** | 100 | 40 | 40 | 40 | 40 | 60 |
| | 13% | | | | | | | |
| | 20% | | | | | | | |
| | 26% | | | | | | | |
| **Macro-oscill.** ±11% | 13% | **Sampling freq (kHz)** | 0.1 | 0.5 | 0.5 | 1 | 1.25 | 1 |

compression level of 13%, using the same set of frequencies as in the previous cases. The details of the experimental protocol are summarised in Table 1, where all compressions refer to the original sample height, while an illustration of the experimental setup and an example of the acquired data are presented in Fig 1.

Using this protocol, three cuboidal samples were employed to investigate the material properties under varying plastisol concentrations: 70%, 80% and 90%, respectively, with 1% w/v trackers.

**Viscoelastic modelling.**    Following the mathematical framework presented in [33], we developed a viscoelastic model to describe the rheological behaviour of the plastisol samples, probed through the mechanical test presented in the previous section.

The vertical strain, $\lambda = \Delta h/h$, can be calculated from the measured height of the sample, $h$, and its variation under compression, $\Delta h$, at any point of the oscillation. Under the assumption of incompressible isotropic material [46], the deformation gradient and the right Cauchy-Green deformation tensor are given by

$$\boldsymbol{F} = \begin{pmatrix} \frac{1}{\sqrt{\lambda}} & & \\ & \frac{1}{\sqrt{\lambda}} & \\ & & \lambda \end{pmatrix}, \qquad \boldsymbol{C} = \begin{pmatrix} \frac{1}{\lambda} & & \\ & \frac{1}{\lambda} & \\ & & \lambda^2 \end{pmatrix}. \tag{15}$$

The stress arising from the applied deformation can be modelled in terms of the second Piola-Kirchhoff (PK2) stress tensor $\boldsymbol{S}$ defined in $\Omega_0$. Again, it is convenient to separate its deviatoric component, $\boldsymbol{S}'$, from a term containing the hydrostatic pressure $P$ and to further decompose $\boldsymbol{S}'$ into the sum of a purely elastic component, $\boldsymbol{S}'_e$, and a viscous one, $\boldsymbol{S}'_v$:

$$\begin{aligned} \boldsymbol{S} &= \boldsymbol{S}' + PJ\boldsymbol{C}^{-1} \\ &= \boldsymbol{S}'_e + \boldsymbol{S}'_v + PJ\boldsymbol{C}^{-1}. \end{aligned} \tag{16}$$

Here, the elastic term captures the stress generated by the total sum of macro- and micro-deformations, while the viscous term accounts for the time-dependent response of the material resulting from the small-amplitude oscillations. An analytical model of $\boldsymbol{S}'_e$ for a hyperelastic material can be obtained from Eq 7 using the relation $\boldsymbol{S}' = \boldsymbol{F}^{-1}\boldsymbol{P}'$ and choosing a specific incompressible strain energy density function $W_e$:

$$\boldsymbol{S}'_e = 2\frac{\partial W_e}{\partial I_{\hat{C}}}\left[ J^{-\frac{2}{3}}\mathbb{1} - \frac{I_{\hat{C}}}{3}\boldsymbol{C}^{-1} \right] + 2\frac{\partial W_e}{\partial II_{\hat{C}}}\left[ J^{-\frac{2}{3}}I_{\hat{C}}\mathbb{1} - J^{-\frac{4}{3}}\boldsymbol{C} - \frac{2II_{\hat{C}}}{3}\boldsymbol{C}^{-1} \right] \tag{17}$$

In the absence of a known strain energy density function capable of capturing the elastic and viscous response of this material, we employed the same model for both the elastic and viscous

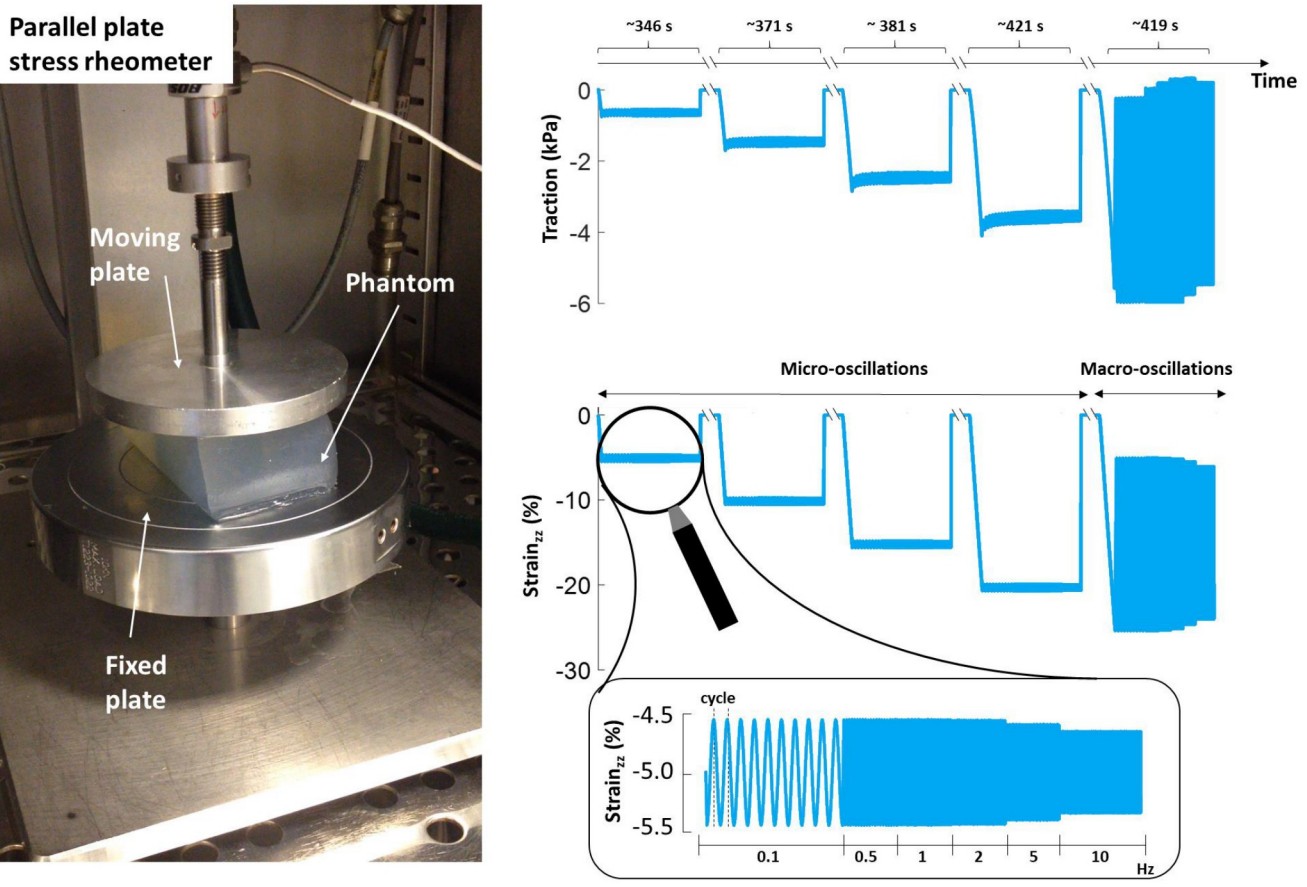

**Fig 1. Viscoelastic characterisation of plastisol material.** The experimental setup used for the viscoelastic characterisation of the phantom material (left). The moving plate was used to apply a fixed vertical compression on the sample and the load was measured through an integrated sensor. For each compression level, the phantom was subjected to four consecutive micro-oscillations and one macro-oscillation test (right). Each of those tests consisted of a series of oscillatory compression cycles centred around the selected compression level at different frequencies, as presented in the zoomed panel. Figure produced after the model proposed in [33].

component of the PK2 stress tensor, with the latter term further subjected to the Caputo fractional time derivative of order $\alpha$ [48]:

$$\mathbf{S}'_e = \mathbf{S}'_e(\boldsymbol{\mu}), \qquad \mathbf{S}'_v = \mathbf{S}'_v(\boldsymbol{\delta}) = D_t^\alpha \mathbf{S}'_e(\boldsymbol{\delta}) \tag{18}$$

Here $\boldsymbol{\mu} = [\mu_1, \mu_2]$ and $\boldsymbol{\delta} = [\delta_1, \delta_2]$ are the sets of material parameters defined in the model. Fractional calculus models have been proposed as a simple and effective way to represent the viscoelastic properties of complex systems like soft tissue [49], and the suitability of this approach was validated on the experimental data. The complete expression of the PK2 stress tensor thus reads [33]:

$$\mathbf{S} = \mu_1 \mathbf{S}'_{e1} + \mu_2 \mathbf{S}'_{e2} + \delta_1 D_t^\alpha \mathbf{S}'_{e1} + \delta_2 D_t^\alpha \mathbf{S}'_{e2} + PJ\mathbf{C}^{-1} \tag{19}$$

where the material parameters have been factored out due to the linearity of the model in the parameters. Given the mild viscosity observed in the hyperelastic material employed to validate the model (see Results), it is relevant to present the case when the fractional derivative order $\alpha$ approaches zero. In this case, $\mathbf{S}'_v$ approaches the same form as its elastic counterpart

[50] and the full $\boldsymbol{S}'$ can be written as

$$\boldsymbol{S}' = \boldsymbol{S}'_e(\mu) + \lim_{\alpha \to 0} D_t^\alpha \boldsymbol{S}'_e(\delta) = \boldsymbol{S}'_e(\mu) + \boldsymbol{S}'_e(\delta). \tag{20}$$

Similar to what done for the PK2 stress tensor, when viscosity is non-negligible, the strain energy density function can be expressed as the linear combination of an elastic and viscous term:

$$W = W_e(\mu; I_{\hat{C}}, II_{\hat{C}}) + W_v(\delta; I_{\hat{C}}, II_{\hat{C}}, \partial_t^\alpha I_{\hat{C}}, \partial_t^\alpha II_{\hat{C}}) \tag{21}$$

As a consequence, by applying the addition rule for limits, also the intrinsic shear modulus presented in Eq 13 can be written as the addition of two components:

$$\begin{aligned}
\mu &= \mu_e + \mu_v \\
&= \lim_{\beta \to 0} 2\left(\frac{\partial W_e}{\partial I_{\hat{C}}} + \frac{\partial W_e}{\partial II_{\hat{C}}}\right) + \lim_{\beta \to 0} 2\left(\frac{\partial W_v}{\partial I_{\hat{C}}} + \frac{\partial W_v}{\partial II_{\hat{C}}}\right).
\end{aligned} \tag{22}$$

For small fractional derivative orders, the same constitutive law can be employed for both $W_e$ and $W_v$, however keeping the scaling parameters $\boldsymbol{\mu}$ and $\boldsymbol{\delta}$. Thus, using the modified Mooney-Rivlin law given in Eq 9, the intrinsic material shear modulus is calculated as

$$\mu = 2(\mu_1 + \delta_1). \tag{23}$$

**Rheological data analysis and model fitting.** The modified Mooney-Rivlin strain energy density function proposed in Eq 9 was used to fit the measured traction to the (3,3) component of the Cauchy stress tensor obtained from the vertical displacement. Specifically, using the relationship [41]

$$\boldsymbol{\sigma} = J^{-1} \boldsymbol{F} \boldsymbol{S} \boldsymbol{F}^T \tag{24}$$

and substituting Eq 16 into Eq 24 it is possible to express the Cauchy stress tensor as a function of the material parameters of the selected hyperelastic law:

$$\begin{aligned}
\boldsymbol{\sigma}(\boldsymbol{\mu}, \boldsymbol{\delta}) &= \boldsymbol{\sigma}'_e(\boldsymbol{\mu}) + \sigma'_v(\boldsymbol{\delta}) + P\mathbb{1} \\
&= J^{-1} \boldsymbol{F} \boldsymbol{S}'_e(\boldsymbol{\mu}) \boldsymbol{F}^T + J^{-1} \boldsymbol{F} \boldsymbol{S}'_v(\boldsymbol{\delta}) \boldsymbol{F}^T + P\mathbb{1}
\end{aligned} \tag{25}$$

To model the vertical normal component of the total Cauchy stress tensor measured by the rheometer ($\sigma_{(3,3)} = \sigma'_{(3,3)} + P$), the hydrostatic pressure must still be evaluated. Since no stress is applied on the free surfaces of the sample, e.g. $\boldsymbol{n}_1 = (1, 0, 0)^T$, then $(\boldsymbol{\sigma}' + P\mathbb{1})\boldsymbol{n}_1$ must be zero. $P$ can then be formulated as

$$P(\boldsymbol{\mu}, \boldsymbol{\delta}) = -[\sigma'_{e(1,1)}(\boldsymbol{\mu}) + \sigma'_{v(1,1)}(\boldsymbol{\delta})] \tag{26}$$

which can be replaced in Eq 25 to obtain the traction applied on the loaded surface, returning an analytical expression of the applied traction:

$$\sigma_{(3,3)} = [\sigma'_{e(3,3)}(\boldsymbol{\mu}) - \sigma'_{e(1,1)}(\boldsymbol{\mu})] + [\sigma'_{v(3,3)}(\boldsymbol{\delta}) - \sigma'_{v(1,1)}(\boldsymbol{\delta})] \tag{27}$$

For each sample, let now $t_{ij}^D$ be the set of traction data measured in the direction of compression at all time points $k = 1, 2, \ldots, N$ of the $j^{th}$ frequency cycle (with $j = 1, 2, \ldots, 6$, as 6 vibration frequencies have been investigated, excluding the data points corresponding to the initial loading of the sample) and $i^{th}$ level of compression (with $i = 1, 2, \ldots, 5$, accounting for the four compression levels chosen for the micro-oscillation tests plus the compression level employed

in the macro-oscillation test). In the same way, let $t_{ij}^M = \sigma_{ij(3,3)}(\tilde{\boldsymbol{\mu}})$ be the modelled traction (from Eq 27), where $\tilde{\boldsymbol{\mu}} = [\boldsymbol{\mu}, \boldsymbol{\delta}]$ is the combination of the free parameters to be estimated, such that $\boldsymbol{\sigma}'(\tilde{\boldsymbol{\mu}}) = \boldsymbol{\sigma}_e'(\boldsymbol{\mu}) + \boldsymbol{\sigma}_\nu'(\boldsymbol{\delta})$. Given the oscillatory nature of the data, the Fourier spectra $\mathcal{F}(t^M)$ and $\mathcal{F}(t^D)$ of the modelled traction and the measured data were employed to define the objective function to minimise. The objective function, representing the distance between the fitting model and the observed data, was designed as follows and minimised through linear least squares regression:

$$\text{error}_\% = 100 \times \sqrt{\sum_{i,j,k} \frac{|\mathcal{F}(t_{ij}^D)_k - \mathcal{F}(t_{ij}^M)_k|^2}{|\mathcal{F}(t_{ij}^D)_k|^2}} \qquad (28)$$

The normalisation factor is such that the fitting error reaches 100% when all free parameters are equal to zero.

Note that $\alpha$, the order of the Caputo fractional derivative, is not part of $\tilde{\boldsymbol{\mu}}$. Instead, the range between $\alpha = 0$ (Hooke's Law) and $\alpha = 1$ (Newton's fluid model) was iteratively investigated, seeking the value that produced the best fit with a ±0.005 precision.

## Experimental validation of analytical framework in phantom

An inflation experiment was devised to validate the mathematical framework. An inclusion inside a soft tissue-mimicking phantom was inflated to different levels, to reproduce the stress generated by an expanding tumour. The pressure applied on the surrounding material was quantified as detailed in the next section. For each inflation state, the shift in shear modulus of the phantom material was reconstructed using MR elastography. Details of the wave generation, image acquisition and $G'$ reconstruction are then presented. High-resolution MR images of the inclusion were also acquired and estimates of the strain fields were obtained using image registration. Given the expected increase in complexity of the wave-behaviour compared to the modelled results, where a simple form of the deformation gradient $\boldsymbol{F}$ was chosen, an improved method to assess the local deformation sensed by the probing waves is finally proposed.

**Inflating pressure measurement.** A CH6 Foley catheter (The Vet Store, Bradford, UK) was inserted into the phantom and inflated with water using a SOFT-JECT®Luer syringe. To provide an access for a tumour-mimicking inclusion, a metallic tube with the diameter of the catheter was inserted in the plastisol matrix during the curing period and gently removed afterwards. To minimise viscous wave attenuation for the MRE data acquisition, the pre-made catheter access was located less than 20 mm away from the side of the phantom in contact with the piston used for the generation of the shear waves.

The pressure applied by the injected water onto the inner wall of the inflated balloon was measured using an MPXV5100DP integrated silicon pressure sensor, connected to the catheter through a series of 3-way tap stopcocks (Fig 2a). The output signal, read through an Arduino®Mega board, returned the average radial stress exerted onto the compliant balloon inner membrane (Fig 2b). To retrieve the actual pressure exerted by the balloon walls on the phantom itself, the pressure required to expand the compliant balloon when no external resistance is applied was characterised and subtracted from the acquired measurements (Fig 2c).

Preliminary experiments showed that injected volumes larger than 0.4 mL could lead to phantom rupture, hence limiting the maximum balloon size to said value. Over a volume range of 0 to 0.4 mL, the balloon was inflated in steps and the pressure was recorded for 5 to 10 min, to allow the time-dependent stress relaxation of all solid components in the system (phantom+balloon) to reach an equilibrium. A more accurate volume estimation was obtained

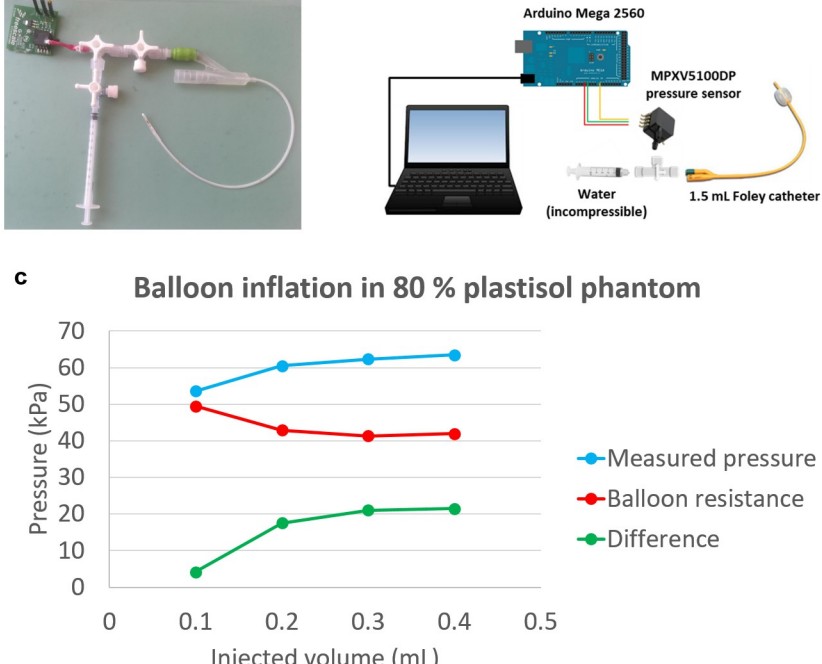

**Fig 2. Details of the inflation experiment.** (a) The Foley catheter used to mimic tumour expansion was connected to a MPXV500DP pressure sensor to measure the radial stress generated through inflation. (b) Schematics of the pressure data acquisition scheme. (c) The actual pressure exerted on the surrounding material is obtained through the subtraction of the intrinsic balloon resistance from the measured pressure.

through the intensity-based segmentation of the balloon from high resolution MR images. The principal axis of the segmented balloon were calculated to assess its dimensions. While one of the principal axes was fixed along the catheter line, the other axes grew in relative magnitude at each inflation state. This was due to the fact that the balloon was not located at the tip but some distance back along the catheter. A ratio of the axis close to 1 indicates a closer approximation of a perfect sphere. A close approximation of a spherical shape was achieved when 0.4 mL of water were injected into the catheter line, as can be seen in Table 2. Because of the particular manner in which the balloon underwent inflation (whereby the equatorial region would bulge out before the poles) the quoted major axis lengths are worst-case estimates, meaning the balloon was likely more spherical than indicated in the table. However, the precise shape could not be determined due to imaging resolution limit. Three replicates of the

**Table 2. Minor-to-major principal axes ratio in the segmented balloon.** The figures correspond to worst-case estimates, as the balloon would typically inflate from the equatorial region at low volumes but the major axis could not be accurately determined due to the limits in imaging resolution.

| Injected volume (mL) | Average ratio (-) |
|---|---|
| 0.0 | 0.25 ± 0.07 |
| 0.1 | 0.33 ± 0.10 |
| 0.2 | 0.50 ± 0.15 |
| 0.3 | 0.74 ± 0.18 |
| 0.4 | 0.85 ± 0.16 |

inflation experiment were carried out; in the last replicate, gadolinium (Gadovist®) was diluted in water to a 0.3 mM concentration to provide better MR contrast.

For the entire duration of each experiment, the time interval during which the balloon was kept inflated at each volume was recorded, so that the pressure acquisition could be repeated one more time on the bench for reproducibility. This was not possible for the second replicate as the balloon popped.

Eq 8 was then used to fit the experimental data. In this case, we assumed that any residual viscoelastic relaxation of the material had reached asymptote at the time of measurement. Consequently, unlike the viscoelastic model employed for the rheological characterisation of plastisol, the purely elastic constitutive equation in Eq 9 was deemed sufficient to describe the material behaviour.

**MR images.** MR images of the phantom were acquired at the different levels of inflation of the balloon catheter after each pressure measurement. During both processes, the phantom was positioned on a supporting stage, located at the centre of the bore of a 3 T Philips Achieva Multi-Transmit scanner (Philips Healthcare, Best, Netherlands), and kept in position by a back wall and top plate with adjustable height. All contact surfaces were coated with lubricant to avoid friction with the supporting stage, that would result in different boundary conditions from those assumed in the analytical model. To generate the shear waves required for elastography imaging, the custom-made electromagnetic transducer presented in [33] was employed. A schematics of the vibration rod used to propagate shear waves through the phantom is shown in Fig 3.

High-resolution MR images of the balloon (Fig 3) were acquired at each inflation state using a spin echo sequence with TR/TE = 1394 ms/ 13 ms (TR = 1239 ms for the third replicate). The image matrix was 144 pixel × 144 pixel, while the number of slices was set equal to 80 or 90, depending on the final size of the phantom, with no gaps. The image resolution was $0.889 \times 0.889 \times 0.89$ mm³. A double SENSE Flex Large coil was used for signal enhancement.

Wave images represent the complex displacement fields generated by the shear waves (Fig 3). They were acquired using a GRE sequence [51] (TR/TE: 187.5 ms/9.21 ms) with motion encoding gradients (MEGs) along the phase (p), frequency (m) and slice (s) encoding directions. MEG and transducer frequency were both set at 210 Hz. 13 sagittal slices with a 2 mm isotropic resolution and no gap were acquired. Each slice covered an in-plane FOV of $128 \times 128$ mm² centred on the inclusion. An imaging plane orthogonal to the direction of the catheter was chosen at the equatorial portion of the balloon where the geometry remained

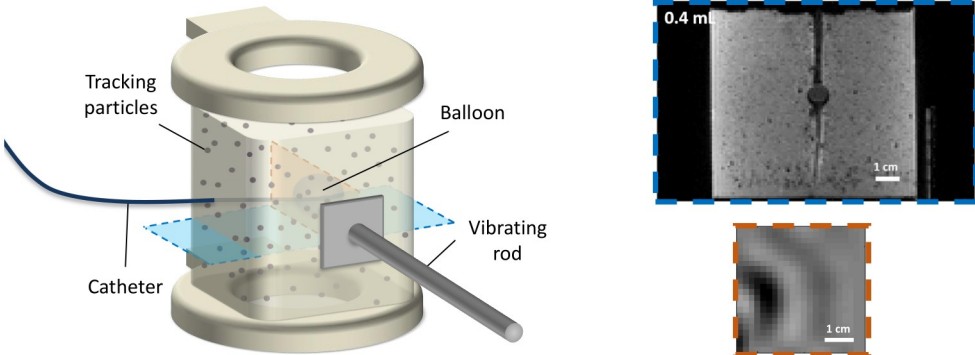

**Fig 3. MR image acquisition.** Schematic representation of the inflated balloon inside the phantom and of the vibrating rod used to generate the shear waves (left). A high-resolution MR image of the inclusion (coronal, blue dotted lines) and a wave image (sagittal, red dotted lines) are also shown (right).

closest to a sphere in all inflation states. This allowed sampling of the mechanical states from where the confounding effects were minimal, and eliminated the catheter line in the image, as well as avoided the polar regions where the resolution limit would increasingly limit the data analysis. 8 time points were sampled along the wave cycle. 4 signal averages were used, leading to a scan time of $\sim 25$ min for each inflation state.

3D maps of the component of the elasticity tensor probed by the shear waves, i.e. $G'$, were reconstructed using a state-of-the-art curl-based approach [52] with a 5 voxel isotropic reconstruction window. Pre-filtering was carried out using a 3D Gaussian kernel ($\sigma = 1.5$ voxel, support = 5 voxel isotropic).

The voxel-wise ratio between the shear modulus reconstructed at each inflation state and that of the deflated case returned the relative shift induced by the deformation. Given the different spatial domains where the data were acquired, the $G'$ map obtained from the deflated balloon domain had to be mapped into each deformed configuration using the macro-deformation fields estimated from high-resolution MR image registration (see next section).

Given the non-uniform location of the wave source throughout different replicates of the inflation experiment, a Cartesian representation of the variation in shear modulus around the inclusion did not offer an easy comparison among experiments or with the corresponding analytical results. A representation based on radial coordinates makes the understanding of the deformation probed by the shear waves simpler, as shown in Fig 4. With this visualisation, 0˚ and ±180˚ lie along the axis parallel to the mean k-vector calculated in the vicinity of the balloon.

**Local deformation probed by shear waves.** The local macro-deformation induced at each inflation state was estimated through the non-rigid registration of the high-resolution MR images using the open-source software SimpleElastix [53]. The registration strategy, based on B-Splines, was previously optimised and validated using a synthetic deformation image series. Among the parameters, the spacing between the nodes of the B-Splines grid, as well as the cost function representing the similarity between the fixed and moving image, were found to be crucial for a successful registration. For the inflation experiment, a 4.45 mm spacing and mutual information were chosen. Improved results were obtained using a multi-resolution approach.

Experimentally, it is difficult to achieve perfect plane waves. The calculation of the gradient of the wave images allowed the voxel-wise estimation of the direction of the wave vector $\mathbf{K}$. A non-uniform wave propagation direction could in fact result in the local sampling of a

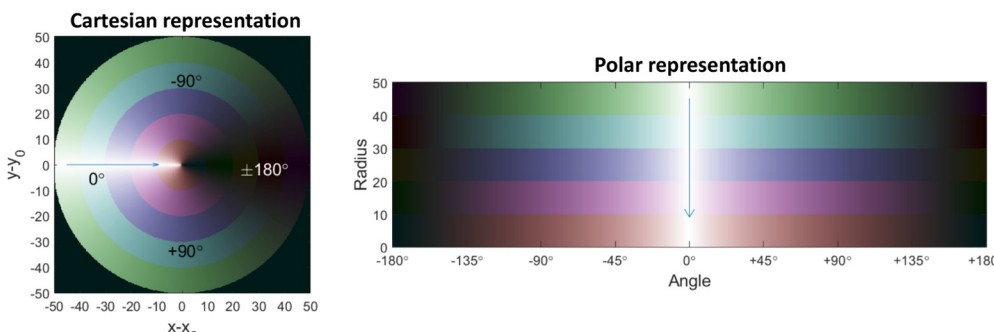

**Fig 4. A non-Cartesian way to visualise the impact of the mimicked tumour-generated strain around the inflated balloon.** Left: example of Cartesian representation of the inclusion where circular regions at different radial distances from the centre of the inclusion are showed with different colours. Right: a better visualisation, independent on the mean direction of propagation of the waves, is obtained through the unravelling of the image following the perimeter of each circular region.

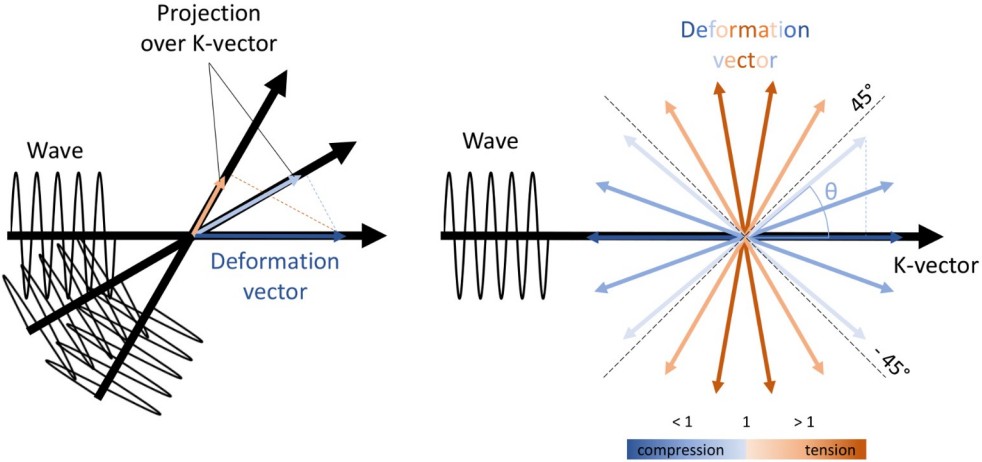

**Fig 5. A new metric to account for the local deformation probed by the shear waves.** Left: shear waves probe the material shear modulus along their direction of propagation, identified by the k-vector. Intuitively, this is comparable to the projection of the deformation field generated by the macro-deformation onto the k-vector. Right: for a spherical deformation, the angle between the deformation vector and the k-vector determines the magnitude of the probed material displacement and allows to differentiate between compression and tension.

component of the deformation vector that is different from the one expected from the mean k-vector. To account for that, we propose a new index, the "probed stretch" *CKK*, defined as

$$CKK = \frac{CK \cdot K}{|K^2|} \qquad \begin{cases} > 1, & \text{extension} \\ < 1, & \text{compression} \end{cases} \tag{29}$$

This scalar quantity represents the projection of the right Cauchy-Green strain tensor on the direction of the k-vector, independent of wave attenuation. It provides an identifier of the local deformation probed by the shear waves, an intuitive representation of which is shown in Fig 5. The probed stretch will be used as a reference metric to determine the ability of the model to correctly identify the regions of compression and tension in the investigated setting, as well as to associate the reconstructed variation in $G'$ to the probed deformation.

## Results

### Analytically derived shear modulus distribution

A plane-wave travelling through the thick-shelled sphere will appear as a $\theta$-polarised $r$-propagating wave or as a $r$-polarised $\theta$-propagating wave in region Ⓐ or Ⓑ around the inclusion presented in Fig 6a. The generated displacements are characterised by

$$Ⓐ \quad \nabla \boldsymbol{u}_{\mathbb{C}}^{r,\theta} = \begin{bmatrix} 0 & 0 & 0 \\ \delta u & 0 & 0 \\ 0 & 0 & 0 \end{bmatrix} \quad \text{and} \quad Ⓑ \quad \nabla \boldsymbol{u}_{\mathbb{C}}^{\theta,r} = \begin{bmatrix} 0 & \delta u & 0 \\ 0 & 0 & 0 \\ 0 & 0 & 0 \end{bmatrix} \tag{30}$$

respectively. In both these two simple cases, the double contraction between the stiffness tensor $\boldsymbol{\mathcal{G}}'$ and $\nabla \boldsymbol{u}_{\mathbb{C}}$, found in the purely elastic version of Eq 3 (following the assumptions of low viscosity made in the modelling of the hyperelastic behaviour of the material and validated in the following section), returns a second-order tensor with only one non-zero component. This result reveals how an ideal plane wave probes only the apparent variation of one single

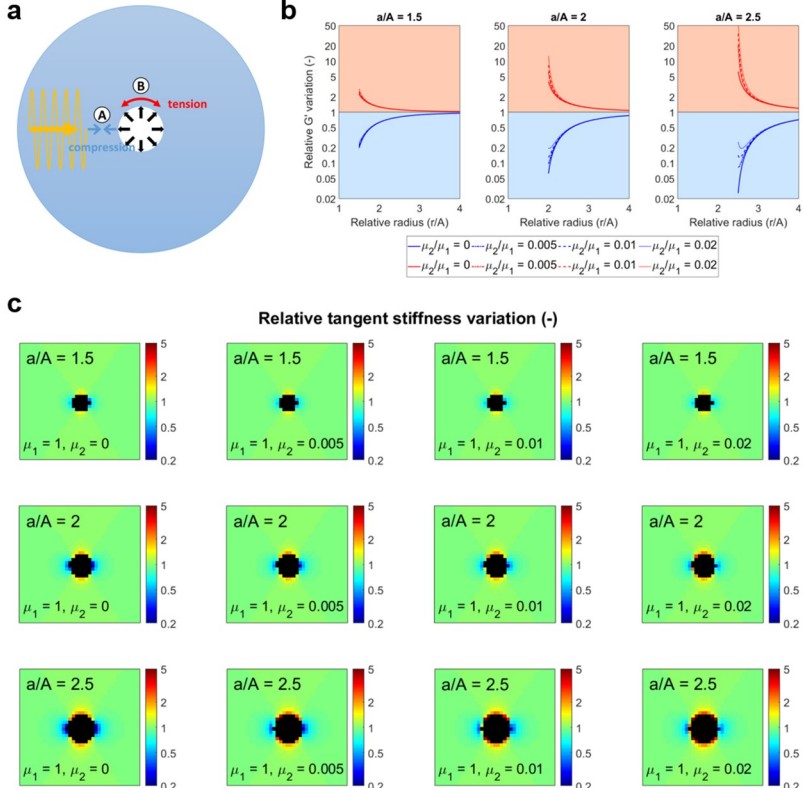

**Fig 6. Modelling the mimicked tumour-generated pressure.** (a) Wave propagation through the thick-shelled sphere. Due to the local spherical coordinate orientation, at Ⓐ, the wave will appear as $\theta$-polarised, travelling in the radial direction. At Ⓑ, the opposite is found. (b) When the waves approach the inner sphere head-on, inflation leads to an apparent softening of the material (blue curves in the blue region). In contrast, along the peripheral interface of the inner sphere, a stiffening is detected (red curves in the red region). Given a constant $\mu_1 = 1$, the parameter $\mu_2$ shapes the curves differently in the two regions, especially in the case of larger deformations (larger $a/A$). The relative $G'$ variation is given in logarithmic scale. (c) 2D representation of the shear modulus variation patterns generated by an inflated inner sphere as probed by plane waves propagating from left to right.

component of $\boldsymbol{\mathcal{G}}'$, namely $G'$. Following the reasoning laid out for the definition of the "probed stretch" $CKK$, the angle between the k-vector and the local direction of the deformation field modulates the intensity and type of change. Fig 6b shows how the apparent modulus softens when a compression is aligned with the wave propagation direction, case Ⓐ of Fig 6a, while it stiffens when tension is sensed, case Ⓑ. The anisotropic patterns generated around an expanded thick-shelled sphere, as a function of the relative radial position at different radial stretches, are instead plotted in Fig 6c. These analytic predictions will be compared with the patterns measured through MRE in a phantom setting in the last section of the Results.

## Viscoelastic characterisation of phantom material

**The proposed model predicts the viscoelastic behaviour of the material.**    The fitting of the rheological data from all tested samples at different plastisol concentrations returned a minimum error below 4% for fractional derivative orders around 0.1, which suggest a mild viscosity. An example of the quality of the best fit is displayed in Fig 7a. Here, the modified Mooney-Rivlin model proves capable of capturing the nonlinear response to the initial loading, as well as the amplitude of both the macro- and micro-oscillations. This was verified against the entire frequency range employed here (0.1 -10 Hz).

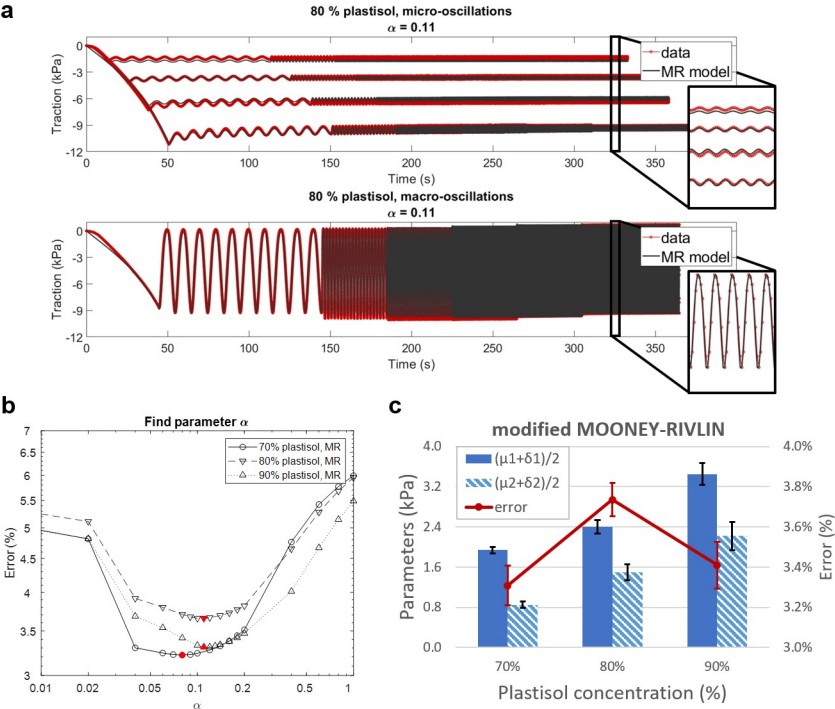

**Fig 7. Modelling the rheological behaviour of the plastisol phantoms.** (a) Example of the best fit of the rheological data acquired from the 70% plastisol sample. Overall the modified Mooney-Rivlin model proved able to reproduce the induced oscillations even at the highest employed frequency. The nonlinear increase in traction was also captured, as well as the relaxation process visible over the explored time-frame. (b) Data from different plastisol concentration are modelled by equally small fractional derivative orders. A fitting error smaller than 4% was obtained for $\alpha$ values between 0.05 to 0.2. The actual minimum, $\alpha_{min}$, is indicated by a filled symbol. (c) Best estimated parameters over fractional derivative orders going from 0.05 to 0.2. Given the parameter coupling, the means of the parameters scaling the linear terms and of those scaling the quadratic ones were considered. The error bars represent the range of values assumed over the selected $\alpha$ range, centred on the average value. The same approach was used for the fitting error.

A similar small error was obtained for $\alpha$ values spanning from 0.04 to 0.2 (Fig 7b). Over this range, the material parameters characterising the elastic and the viscous part of the model appeared linearly coupled, such that the sums $\mu_1 + \delta_1$ and $\mu_2 + \delta_2$ remained constant. Consequently, rather than the best estimate of the single parameters, it was preferred to consider the means $(\mu_1 + \delta_1)/2$ and $(\mu_2 + \delta_2)/2$ and to average their values over the selected $\alpha$ range. The same approach was used for the fitting error. These combined parameters are presented in Fig 7c and are reported in Table 3.

**Plastisol concentration increases material stiffness and nonlinearity.** The loading curve of each sample used for the rheological experiment demonstrated a correlation between plastisol concentration and material nonlinearity (Fig 8a). The coefficient scaling the quadratic term

**Table 3. Best estimate of material parameters characterising the rheological properties of samples made with different plastisol concentrations.**

| Model | Plastisol | $\frac{\mu_1+\delta_1}{2}$ | $\frac{\mu_2+\delta_2}{2}$ | Error$_\%$ |
|---|---|---|---|---|
| Mooney-Rivlin | 70% | 1953 kPa | 872 kPa | 3.22% |
| | 80% | 2320 kPa | 1648 kPa | 3.66% |
| | 90% | 3381 kPa | 2358 kPa | 3.31% |

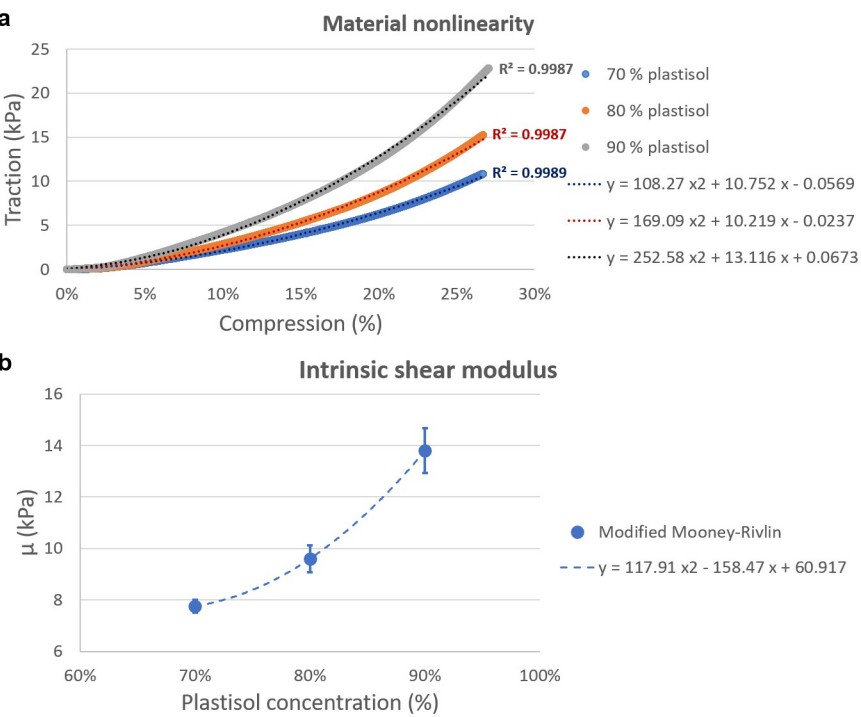

**Fig 8. Material stress-strain nonlinearity.** (a) The stress-strain response of samples made with different plastisol concentrations was fitted with a second-order polynomial function. The increasing value of the parameter scaling the quadratic term associates a higher plastisol content to an increased material nonlinearity. (b) The intrinsic shear modulus of the material was also found to increase with the amount of plastisol in the mixture, following a quadratic trend.

of the polynomial fit, in fact, showed a direct dependence on the plastisol percentage in the mixture.

In a similar way, the material stiffness was found to grow with the plastisol content. Fig 8b showed that the intrinsic shear modulus of the samples, calculated from the modelling parameters according to Eq 23, followed a quadratic trend.

### Fitting analytical model to pressure data

The family of pressure curves obtained from Eq 8 exploring the use of different material parameters $\mu_1$ and $\mu_2$ is shown in Fig 9a. In the absence of $\mu_2$, the model was reduced to the Neo-Hookean case, with the parameter $\mu_1$ working as a scalar and without modifying the trend of the inflating pressure. The quadratic term, instead, controlled the magnitude of the generated pressure at higher levels of inflation. The presence of both parameters gave rise to a qualitative change in pressure-radius relation, namely producing S-shape curves.

The experimentally measured pressure applied by the inflated balloon onto the surrounding phantom is listed in Table 4, together with the corresponding radial stretch ($a/A$). The mathematical formulation proposed in Eq 8 was fitted to the data points obtained from three replicates of the inflation experiment (Fig 9b), showing a correlation between the analytical predictions and the experimental data ($r^2 = 0.87$). Using Eq 14, the best fitting parameters returned an estimate for the intrinsic shear modulus of the material of 11.1 kPa, a good approximation of the mean value of 12.0 ± 0.4 kPa measured through MRE when the balloon was deflated.

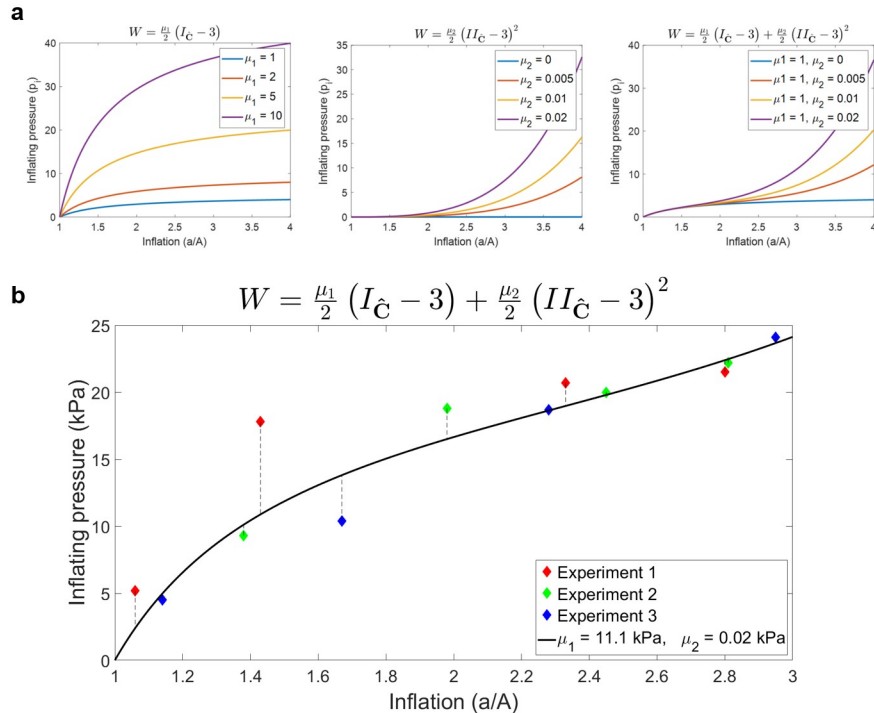

**Fig 9. Fitting the experimental pressure data.** (a) Example family of curves representing the inflating pressure obtained from the linear (left) and quadratic (centre) term of the modified Mooney-Rivlin law employed. The combined contributions generate a characteristic family of S-shaped curves (right). (b) The modified Mooney-Rivlin model provides a good approximation of the pressure data obtained from the three replicates of the inflation experiment ($r^2 = 0.86$).

## Experimental $G'$ patterns agree with analytical predictions

Fig 10 provides the comparison between the probed stretch (first column) and the measured (second column) and analytically predicted variation in $G'$ associated to different inflation levels with respect to the deflated case.

The relative $G'$ variation patterns generated from the thick-shelled sphere approximation matched the alternation between regions of tension and compression estimated through the probed stretch (*CKK*) using the measured displacement. Despite the simplifying assumptions, this qualitative agreement confirms the reliability of the proposed model to associate the correct change in $G'$ to the probed deformation in the investigated setting. However, given the different information portrayed by the probed stretch and the analytical predictions, a quantitative comparison is not possible. Nevertheless, the increase in absolute pixel intensity with higher inflation, visible in both cases, supports the direct relationship between the two results.

**Table 4. Inflating pressure (kPa) / radial stretch (-) measured from the three replicas of the inflation experiment.**

| Inflation | Experiment 1 | Experiment 2 | Experiment 3 |
|---|---|---|---|
| 0.1 mL | 5.2 ± 1.0 kPa / 1.06 ± 0.19 | 9.3 kPa / 1.38 ± 0.23 | 4.5 ± 0.7 kPa / 1.14 ± 0.40 |
| 0.2 mL | 17.8 ± 0.2 kPa / 1.43 ± 0.20 | 18.8 kPa / 1.98 ± 0.26 | 10.4 ± 0.6 kPa / 1.67 ± 0.62 |
| 0.3 mL | 20.7 ± 0.3 kPa / 2.33 ± 0.51 | 20.0 kPa / 2.45 ± 0.28 | 18.7 ± 4.6 kPa / 2.28 ± 0.78 |
| 0.4 mL | 21.5 ± 0.0 kPa / 2.80 ± 0.43 | 22.2 kPa / 2.81 ± 0.35 | 24.1 kPa / 2.95 ± 1.01 |

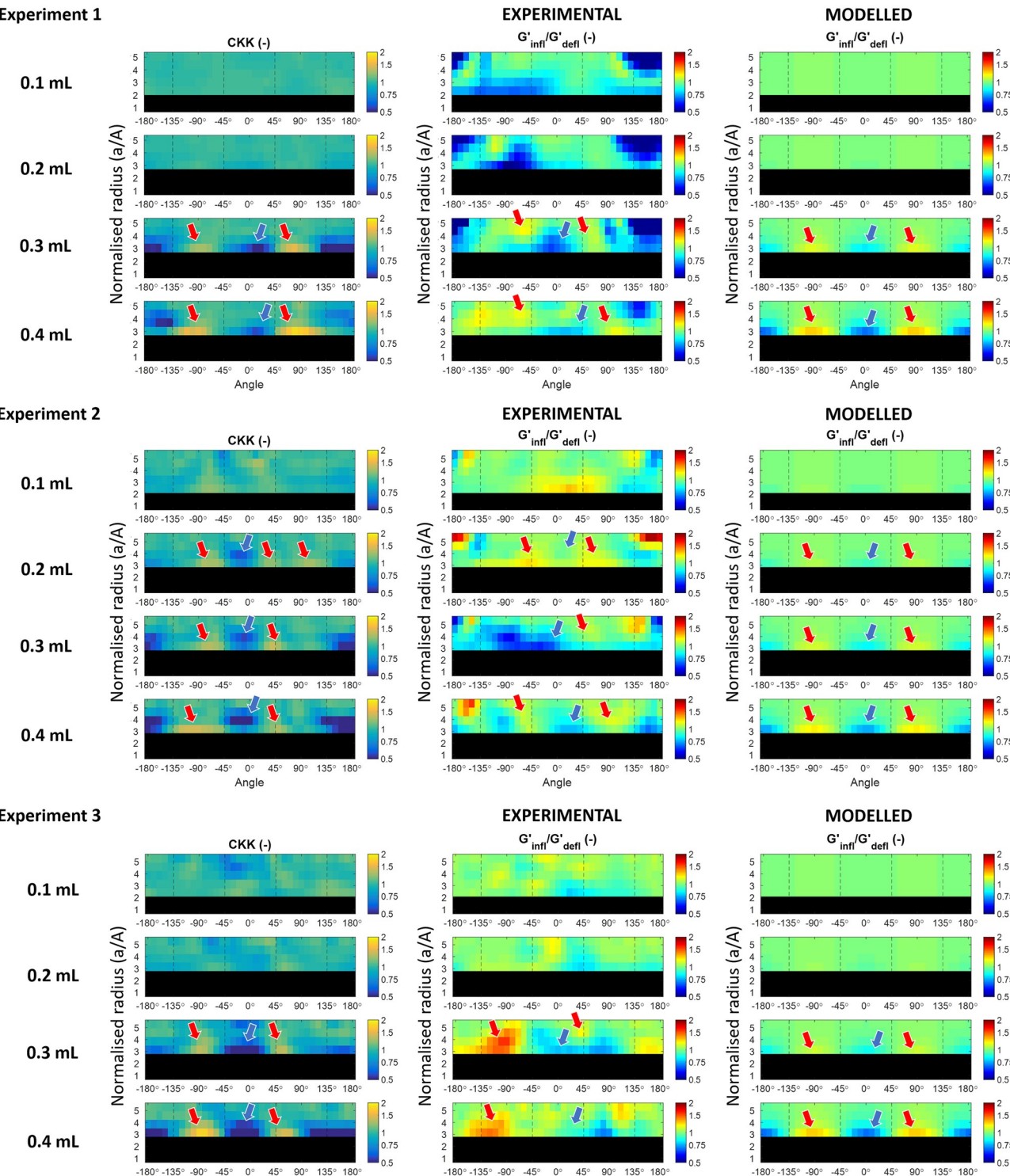

**Fig 10. Measured $G'$ variation supports mathematical model.** Polar representation of the probed deformation (left), calculated using the local k-vector estimated from the wave images, compared with experimental (middle) and analytical (right) relative shear modulus variation at the four inflation levels investigated through three replicas of the inflation experiment. The common elements in the generated patterns are highlighted with arrows. The balloon region is blacked out.

The variation in $G'$ reconstructed from the MRE data also exhibits several elements in common with the $G'$ variation obtained with analytic idealisation (see arrows), especially in the -135˚ to 135˚ region, corresponding to the leading edge of the inclusion. Over these areas, the patterns show a quantitative agreement with the analytical predictions, too. Nevertheless, unexpected patterns are also present, principally at lower strains, where the less prominent nonlinear tissue response results in lower SNR. Furthermore, a better agreement was generally found starting from one or two voxels away from the interface between the inclusion and the surrounding phantom material.

## Discussion

In this paper, the apparent shift in shear modulus of a soft tissue phantom around a pressurised inclusion was investigated using MRE. An analytical framework incorporating the impact of a large-strain deformation on the wave equation for a viscoelastic body was presented, highlighting the dependence of the stiffness tensor $\mathcal{G}^*$ on the underlying strain and on the stress-strain relationship. For the derivation of a mathematical model, the idealised case of an expanding nonlinearly-elastic thick-shelled sphere was used to simulate the pressure exerted by an expanding tumour onto its surroundings. Actual tumours can although present rather irregular shapes, with pronounced eccentricity and/or invasive protrusions that can favour invasion and migration [54]. Some studies have attempted to model a solid tumour using more complicated shapes [55–58], however an axisymmetric geometry is a common starting point in most studies, due to the simpler analytical formulation [4, 8, 59–63]. Here we chose to employ the simplifying assumption, in order for the coupled tumour mechanics and MRE to remain analytically tractable, as well as to establish baseline results against which secondary geometric effects can be benchmarked.

Despite having based the model on an axisymmetric inflation, the employed balloon-catheter did not present a spherical shape until 0.4 mL of water were injected. Nonetheless, the two principal axes orthogonal to the direction of the shaft of the catheter showed comparable values at all inflation states, confirming a symmetric expansion in the plane transverse to the catheter. Consequently, the position of the transducer and the imaging plane were chosen such that we could observe the impact of this circular stretch on $G'$ probed by shear wave propagating in the same plane. The mechanical effects were also sampled from selected regions where the confounding factors were minimal, e.g.: where the signal-to-noise (or effect-to-noise) was expected to be maximum, at the same time avoiding the resolution limit at the polar region that increasingly hamper the analysis. In spite of the experimental set up not perfectly reproducing the conditions used for the definition of model, analogous patterns were obtained between the modelled and reconstructed $G'$ distribution, suggesting that different geometrical conditions might produce comparable results. Another analytical work produced similar pressure-strain curves using different simulated tumour shapes, supporting our simplifying assumptions and mitigating the practical limitations encountered in the design of suitable experimental conditions [40].

Another important assumption which underlies the proposed model concerns the chosen hyperelastic law. While the initially presented wave equations (Eq 3) assumed a full viscoelastic framework, a nonlinear elastic constitutive equation was instead used when describing the apparent shear modulus probed by plane waves when propagating through an inflated thick-shelled sphere. As introduced earlier, the choice to neglect the viscous terms in the latter formulation was dictated by the assumption that MRE was performed at steady state, after any time-dependent deformation had occurred. This allowed to make the analytical modelling more manageable and to better represent the experimental conditions.

With this premise, we could identify a signature anisotropic $G'$ pattern around the expanding inner sphere, using a modified Mooney-Rivlin constitutive equation. This variation in $G'$ displayed a close agreement with the newly introduced "probed stretch", confirming the ability of the developed model to predict the regions of probed tension and compression. This was true despite the experimental challenges to satisfy the plane wave assumption and in spite of the simplified geometric idealisation. The probed stretch, hence, presents itself as an alternative trend indicator, potentially simpler than $G'$ ratio and independent of MRE reconstruction. In a similar way, the strain-generated shift in $G'$ estimated experimentally from the MRE data showed several elements in common with the corresponding analytical predictions. Nevertheless, in this case, a full qualitative correspondence was not observed, especially at lower levels of inflation, where reduced SNR played a significant role in stiffness reconstruction. In light of the agreement between the model and the probed deformation, the dissimilarities can be ascribed to issues in the MRE reconstruction process. The assumption of local homogeneity required in the curl-based approach was not respected at the interface between the balloon and the surrounding phantom, condition that has been shown *in silico* to lead to the underestimation of the reconstructed $G'$ [64]. A recent reconstruction method based on localised divergence-free FEM has shown to better handle material discontinuities and could produce improved results [64]. The assumption of near-incompressibility of tissue made for the employed reconstruction and generally accepted for many MRE reconstructions can also impact $G'$ estimation. While the discussion of the best reconstruction method for the presented investigation falls outside the scope of this paper, a complete review of the limitations of the most common reconstructions used in MR elastography, including the one used here, can be found in [40]. The liquid content of the balloon provides another challenge for the MRE reconstruction, as it does not support the propagation of shear waves. In the absence of shear displacement, noise in the MR images would be interpreted as high-frequency small-amplitude displacement and associated to a very soft material. Given the size of the reconstruction window, this underestimation can affect the estimation of $G'$ for the voxels close to the discontinuity. Despite being limiting factors in the developed experimental settings, such issues are not expected to be as relevant in the case of a real tumour, as the material transition would be less abrupt and the tumour would be capable of supporting shear wave propagation. On the other hand, more complex strain and wave propagation patterns can be anticipated. Their investigation and incorporation into the analytical simulation could provide a better understanding of the additional challenges encountered *in vivo*.

The predicted association of extension/compression to material stiffening/softening was comparable with a previous analysis where the uni-axial load of a Neo-Hookean incompressible material was investigated [33]. Consistently, the magnitude of variation in apparent modulus seen using the modified Mooney-Rivlin model was found to be less evident in the compression case, especially when the quadratic term becomes more prominent (Fig 6b). The simpler Neo-Hookean model ($\mu_2 = 0$) produced similar patterns (Fig 6c). Nevertheless, this model is incapable of capturing the nonlinear rheology of the material under large strains typical of soft tissue [65]. Exponential constitutive equations have been shown to well represent the stress-strain response of breast tissue [66]. Nevertheless, to keep the mathematical formulation of the inflating pressure in Eq 8 tractable, only the use of the proposed polynomial model was pursued.

The choice of a hyperelastic material was fundamental to generate a strain-associated variation in shear modulus capable of producing a nonlinear stress-strain response The rheological characterisation revealed a direct dependence of the material nonlinearity on plastisol concentration, which reflects the findings of [45]. The intrinsic shear modulus of the material was found to be directly dependent on the plastisol concentration also (Fig 8b). For the three

investigated plastisol concentration, $\mu$ was estimated to grow from 7.8 kPa to 13.8 kPa, significantly above the 0.5 to 3 kPa measured in healthy liver [67], breast [68] or brain tissue [69] *in vivo* using MRE. In order to keep the sample stiffness comparable with that of soft tissue and, at the same time, to achieve a sufficient nonlinear response in the inflation experiment, a trade-off was chosen using a plastisol concentration of 80%.

The material was also found to be characterised by a mild viscosity. This was suggested by the low fractional derivative orders that produced the best fit of the rheological data at the investigated frequencies. The low measured viscosity justifies the purely elastic assumption used for the thick-shelled sphere approximation. At the same time, the small $\alpha$ values can explain the parameter coupling encountered while fitting the rheological data. Since the same model was used to describe both the elastic and viscous part of the deviatoric PK2 stress tensor, with the latter being subjected to fractional derivative (see Eq 18), $S'_v$ approaches the same form as its elastic counterpart when the fractional derivative order approaches zero, as shown in Eq 20. This makes the parameters $\mu$ and $\delta$ redundant. The same reasoning supports the analytical estimate of the intrinsic shear modulus of the material proposed in the last section of the Analytical Model.

Here we showed that knowledge of the deformation gradient, under the choice of a specific constitutive equation, can be used to predict the associated shift in shear modulus as probed by the shear waves in the case of an axisymmetric inflation. Conversely, a known deformation field can be incorporated in the inverse solution of the wave equation, hence allowing to effectively "undo" the apparent anisotropy effect and to reconstruct the intrinsic shear modulus of the material in the undeformed case. This was recently demonstrated in PVA samples under a simpler setting of uni-axial compression [33]. Fovargue *et al.* have then applied the idea by "undoing" the $G'$ anisotropy measured through MRE using a phenomenological formulation in the case of radial expansion, both in phantoms and *in vivo* [40, 70]. The present work therefore sets the theoretical ground necessary to bridge these studies, providing a first general model that describes the signature shear modulus pattern generated by a spherical growth which can be detected by MRE. Furthermore, our model provides the means to be able to retrieve the pressure acting on the tissue once the level of inflation is known, thereby enabling the investigation of tumour forces from imaging data. While the results presented herein serve as a starting point to elucidating the utility of tumour-generated forces in MRE quantification, further work will be required when extending the experimental validation to real tissue. Employment of the proposed mathematical framework within an *in vivo* setting will likely require involving the different types of solid and fluid tumour-related stresses outlined in the introduction. At the same time, some of the experimental limitations encountered here (such as the presence of the catheter access hole and the use of a water-filled balloon impenetrable to shear wave propagation) will be alleviated. As a glimpse into the future, a first clinical application covering 15 malignant breast lesions and 1 breast fibroadenoma correlated an elevated tumour pressure to lymphovascular invasion, using the measured anisotropy of $G'$ around the imaged tumour [70]. Progresses in the model development will enable enhanced MRE reconstruction strategy, facilitating the search of a non-invasive biomarker to gauge metastatic propensity.

## Conclusion

Modelling investigations of tumour-generated strain are sparsely found in the literature, and understanding the impact of the underlying stress on elastographic reconstruction remains an emerging field at present. In this article we have developed an analytical framework to describe the apparent variation in shear modulus generated by an axisymmetric deformation of a

nonlinear viscoelastic material, as probed by shear waves. In this signature pattern of modulus, the magnitude of the deviation from a homogeneous distribution is directly linked to the underlying stress-strain state and its quantification was shown to depend on the chosen material law. The analytic model was tested under a phantom setting and produced $G'$ distributions around the inclusion that was comparable to those observed experimentally. An alternative indicator of the probed stretch, independent of the MRE reconstruction and based on the local direction of shear waves with respect to the tissue deformation, was proposed. Given the difficulty in generating plane waves experimentally, this metric aimed at estimating the tension/compression experienced locally by the shear waves, which showed a better agreement with the analytical predictions. Our findings underline the need to account for the apparent anisotropy in $G'$ caused by the underlying macro-deformation in MRE reconstruction. To this end the presented model takes a first step towards the ultimate application for tumour-generated forces in MRE. To achieve this, further validation in real tissue and developments in both tumour-tissue modelling as well as nonlinear MRE reconstruction will be required.

## Supporting information

**S1 Appendix. Linearised elastic wave equations in the presence of a macroscopic deformation.**
(ZIP)

## Acknowledgments

The authors would like to thank Prof Hazel Screen and her staff at the Institute of Bioengineering of Queen Mary University London (London, UK) for their availability and willingness to let us carry out our rheological tests with their instrumentation.

## Author Contributions

**Conceptualization:** Ralph Sinkus, Jack Lee.

**Formal analysis:** Marco Fiorito, Daniel Fovargue.

**Funding acquisition:** David Nordsletten, Ralph Sinkus.

**Investigation:** Marco Fiorito, Myrianthi Hadjicharalambous.

**Methodology:** Marco Fiorito, Adela Capilnasiu, Myrianthi Hadjicharalambous.

**Project administration:** Marco Fiorito, Jack Lee.

**Supervision:** Ralph Sinkus, Jack Lee.

**Validation:** Marco Fiorito.

**Visualization:** Marco Fiorito.

**Writing – original draft:** Marco Fiorito, Jack Lee.

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
