## [Decision Letter · Decision Letter 0]

5 Mar 2021

PONE-D-21-02832

Modelling the mechanical signature of pressurised tumours on surrounding tissue and its impact in MR elastography

PLOS ONE

Dear Dr. Fiorito,

Thank you for submitting your manuscript to PLOS ONE. After careful consideration, we feel that it has merit but does not fully meet PLOS ONE’s publication criteria as it currently stands. Therefore, we invite you to submit a revised version of the manuscript that addresses the points raised during the review process.

We look forward to receiving your revised manuscript.

Kind regards,

Jose Manuel Garcia Aznar

Academic Editor

PLOS ONE

Journal Requirements:

2) We note that you have included the phrase “data not shown” in your manuscript. Unfortunately, this does not meet our data sharing requirements. PLOS does not permit references to inaccessible data. We require that authors provide all relevant data within the paper, Supporting Information files, or in an acceptable, public repository. Please add a citation to support this phrase or upload the data that corresponds with these findings to a stable repository (such as Figshare or Dryad) and provide and URLs, DOIs, or accession numbers that may be used to access these data. Or, if the data are not a core part of the research being presented in your study, we ask that you remove the phrase that refers to these data.

3) Please include captions for your Supporting Information files at the end of your manuscript, and update any in-text citations to match accordingly. Please see our Supporting Information guidelines for more information: http://journals.plos.org/plosone/s/supporting-information.

Reviewers' comments:

Reviewer's Responses to Questions

**Comments to the Author**

1. Is the manuscript technically sound, and do the data support the conclusions?

Reviewer #1: Yes

2. Has the statistical analysis been performed appropriately and rigorously? 

Reviewer #1: N/A

3. Have the authors made all data underlying the findings in their manuscript fully available?

Reviewer #1: Yes

4. Is the manuscript presented in an intelligible fashion and written in standard English?

Reviewer #1: Yes

5. Review Comments to the Author

Reviewer #1: The paper presents a formulation to retrieve the mechanical state of the surrounding tissue of a growing tumour. They used an experimental set-up in which a plastisol simulated the surrounding soft tissue of the tumour and the tumour was replaced by a sphere incision in which they applied different pressure levels. The strain dependent apparent shear modulus of the surrounding tissue (plastisol) is recovered with MR-Elastography and compared against analytical results.

The topic is interesting, however, the authors should make an effort to highlight the new contribution of this paper against their previous work (Capilnasiu et al., 2019) and to discuss the validity of the model to mimic a real growing tumour.

Comments:

1. I would recommend to review the title, authors present a general formulation of finite strains and apply it to determine the mechanical state of a non-living material in which a defect with the shape of a sphere is introduced and them a pressure inserted. There are few reference to the tumour in the main body of the text.

2. The authors perform a theoretical study which is far from mimic the real environment of a growing tumour, I would recommend to rephrase or remove the sentences related to the early application of the model to predict the mechanical state of a real tumour (for exaple the lass sentence of the abstract “a significant step …. treatment efficacy”. In fact, they do not perform any validation neither quantitative nor qualitative with real tissue data.

3. The authors present a complete analytical formulation for finite strains which they introduced in a previous work (Capilnasiu et al., 2019). In my opinion the authors should explain the new contributions in further detail and move the sections related to the general formulation already published to supplementary material. In fact, it is hard to see the novelty of this formulation against their previous work and the analytical part related to the sphere could be further elaborated.

4. Lines 14-15. The authors assume the tumour is a perfect sphere. This is a rather crude assumption in a real tumour growing in a very heterogeneous environment in the human body please at least state the conditions to consider the tumour a perfect sphere.

5. The use of the name “a” for different variables is confusing (internal spatial radius page 5 and 6, subscript equation (21), and shear deformation line 158) please rename the different variables with different names.

6. Line 140. The authors are assuming the “sigma_rr” in the external radius (r=b) is zero, however it is surrounded by additional tissue, please, justify this assumption.

7. Table 1, lines 196-206. Even though, it is stated in the main text (line 205), I think it would be good to include also in the table legend if the percentage of compression in the micro scale refer to a percentage of the macro scale deformation 13% or the total deformation of the sample.

8. Figure 1, it is the same as figure 2 in Capilnasiu et al. (2019), Just the number change, please try to change the figure a little bit to make it clearer.

9. Lines 298-299. “the non-spherical shape of the balloon when not fully inflated”. Please, specify the moment from which the balloon could be considered inflated, and the implications in your formulation since you are considering a perfect sphere.

10. Lines 311-313. You introduce a complete viscoelastic model however them you just use the elastic part. This is really confusing, please disscuss in the Discussion Section.

11. Figure 3. The dimensions of the inclusion sphere compare to the dimensions of the paths for catheter and other devices inclusions are of the same order of magnitude. Please, comment on that and the consequences in the formulation.

12. In the conclusion authors state that “tumour- generated stress has been investigated” however in the discussion section there is just a brief discussion of two previous work related to tumour-generated stress. Authors centre the discussion on the validation of the analytical model with their experiments on plastisol. Authors should further discuss on the limitations on their model to extrapolate results to real tumours and the advance of his work against what has already been published in tumour growth.

Typos:

-Equation (16), “-T” Should be a superindex.

-Page 5, line 129. “we denote”

-Figure 6 Legend, line 5. “blue”

Capilnasiu, A., Hadjicharalambous, M., Fovargue, D., Patel, D., Holub, O., Bilston, L., Screen, H., Sinkus, R., & Nordsletten, D. (2019). Magnetic resonance elastography in nonlinear viscoelastic materials under load. Biomechanics and modeling in mechanobiology, 18(1), 111–135. https://doi.org/10.1007/s10237-018-1072-1

6. PLOS authors have the option to publish the peer review history of their article (what does this mean?). If published, this will include your full peer review and any attached files.

Reviewer #1: No

---

## [Author Response · Author response to Decision Letter 0]

21 Apr 2021

Dear Editor, dear Reviewer,

please refer to the uploaded document "Response to Reviewers.pdf" for the response to your comments.

---

## [Decision Letter · Decision Letter 1]

14 Jun 2021

Impact of axisymmetric deformation on MR elastography of a nonlinear tissue-mimicking material and implications in peri-tumour stiffness quantification

PONE-D-21-02832R1

Dear Dr. Fiorito,

We’re pleased to inform you that your manuscript has been judged scientifically suitable for publication and will be formally accepted for publication once it meets all outstanding technical requirements.

Kind regards,

Jose Manuel Garcia Aznar

Academic Editor

PLOS ONE

Additional Editor Comments (optional):

Reviewers' comments:

Reviewer's Responses to Questions

**Comments to the Author**

1. If the authors have adequately addressed your comments raised in a previous round of review and you feel that this manuscript is now acceptable for publication, you may indicate that here to bypass the “Comments to the Author” section, enter your conflict of interest statement in the “Confidential to Editor” section, and submit your "Accept" recommendation.

Reviewer #1: All comments have been addressed

2. Is the manuscript technically sound, and do the data support the conclusions?

Reviewer #1: Yes

3. Has the statistical analysis been performed appropriately and rigorously? 

Reviewer #1: N/A

4. Have the authors made all data underlying the findings in their manuscript fully available?

Reviewer #1: Yes

5. Is the manuscript presented in an intelligible fashion and written in standard English?

Reviewer #1: Yes

6. Review Comments to the Author

Reviewer #1: In this revised manuscript, the authors have addressed my concerns.

The authors should just correct a typo:

-Line 596 "[did we show this?] "

7. PLOS authors have the option to publish the peer review history of their article (what does this mean?). If published, this will include your full peer review and any attached files.

Reviewer #1: No

---

## [Editor Report · Acceptance letter]

28 Jun 2021

PONE-D-21-02832R1 

Impact of axisymmetric deformation on MR elastography of a nonlinear tissue-mimicking material and implications in peri-tumour stiffness quantification 

Dear Dr. Fiorito:

I'm pleased to inform you that your manuscript has been deemed suitable for publication in PLOS ONE. Congratulations! Your manuscript is now with our production department. 

Kind regards, 

on behalf of

Dr. Jose Manuel Garcia Aznar 

Academic Editor

PLOS ONE